# PLGA-PEG-PLGA hydrogel with NEP1-40 promotes the functional recovery of brachial plexus root avulsion in adult rats

Wenlai Guo[1], Bingbing Pei[2], Zehui Li[1], Xiao Lan Ou[1], Tianwen Sun[1] and Zhe Zhu[1]

[1] Department of Hand Surgery, The Second Hospital of Jilin University, Chang chun, Jilin, China
[2] Department of Orthopedics, Chinese People's Liberation Army Joint Logistics, Support Unit 964 Hospital, Chang chun, Jilin, China

## ABSTRACT

Adult brachial plexus root avulsion can cause serious damage to nerve tissue and impair axonal regeneration, making the recovery of nerve function difficult. Nogo-A extracellular peptide residues 1-40 (NEP1-40) promote axonal regeneration by inhibiting the Nogo-66 receptor (NgR1), and poly (D, L-lactide-co-glycolide)-poly (ethylene glycol)-poly (D, L-lactide-co-glycolide) (PLGA-PEG-PLGA) hydrogel can be used to fill in tissue defects and concurrently function to sustain the release of NEP1-40. In this study, we established an adult rat model of brachial plexus nerve root avulsion injury and conducted nerve root replantation. PLGA-PEG-PLGA hydrogel combined with NEP1-40 was used to promote nerve regeneration and functional recovery in this rat model. Our results demonstrated that functional recovery was enhanced, and the survival rate of spinal anterior horn motoneurons was higher in rats that received a combination of PLGA-PEG-PLGA hydrogel and NEP1-40 than in those receiving other treatments. The combined therapy also significantly increased the number of fluorescent retrogradely labeled neurons, muscle fiber diameter, and motor endplate area of the biceps brachii. In conclusion, this study demonstrates that the effects of PLGA-PEG-PLGA hydrogel combined with NEP1-40 are superior to those of other therapies used to treat brachial plexus nerve root avulsion injury. Therefore, future studies should investigate the potential of PLGA-PEG-PLGA hydrogel as a primary treatment for brachial plexus root avulsion.

Corresponding authors
Tianwen Sun, suntianwen@jlu.edu.cn
Zhe Zhu, zhuzhe1983@jlu.edu.cn

## INTRODUCTION

Brachial plexus avulsion (BPA) is a commonly observed injury in young adults and is often caused by severe trauma, such as that associated with car accidents (*Faglioni et al., 2014*). After BPA, the nerve root is torn from the spinal cord, which causes loss of sensory and motor functions in the corresponding area of innervation. The initial insult and secondary damage result in widespread neuronal death. Unfortunately, nerve regeneration is difficult because of various inhibitory factors, and recovery is almost impossible without surgical treatment. Further, simple surgical treatment often leads to unsatisfactory

results that are accompanied by a high disability rate (*Carlstedt, 2008*; *Gibon et al., 2016*; *Lang et al., 2005*).

Replantation of the avulsed nerve root *in situ* is the ideal treatment for BPA. Replantation can not only restore the anatomical structure and avoid injuries caused by other reconstructive interventions (*Eggers et al., 2010*; *Limthongthang et al., 2013*; *Shin et al., 2005*), but also greatly reduces post-injury neuralgia (*Ciaramitaro et al., 2017*; *Teixeira et al., 2015*; *Zhou et al., 2017*). Nevertheless, various inhibitory factors produced in post-injury nerve tissue, limit neural regeneration and result in poor functional recovery (*So & Xu, 2015*). To promote the recovery of nerve function, more attention has been given to the promotion of axon regeneration or the elimination of inhibitory factors associated with nerve damage.

Nogo receptors (NgRs), including NgR1, NgR2, and NgR3, are endogenous myelin-associated protein-related receptors in the central and peripheral nerves (*Chen et al., 2000*; *Young, 1996*). Nogo-A, the ligand with the highest affinity for NgR1, can bind to NgR1, inhibit axonal growth, resulting in growth cone collapse. Nogo-A extracellular peptide residues 1-40 (NEP1-40), the most common NgR1 antagonist, comprises a small polypeptide that competes with NgR1 by targeting Nogo-66, the main inhibitory region of Nogo-A (*GrandPré, Li & Strittmatter, 2002*). Therefore, NEP1-40 can relieve the neurosuppressive effect of Nogo-A (*Cao et al., 2008*; *Chen et al., 2000*; *GrandPré, Li & Strittmatter, 2002*; *Kempf & Schwab, 2013*; *Xu et al., 2017*), increase the expression of growth-associated protein 43 (GAP-43) and microtubule-associated protein 2 (MAP-2) in nerve tissue, reduce the levels of amyloid-β (A4) precursor protein (APP), and, consequently, facilitate nerve regeneration, axonal growth as well as functional recovery (*Li & Strittmatter, 2003*; *Mingorance et al., 2006*; *Wang et al., 2007*; *Xu et al., 2017*).

BPA can (i) cause nerve tissue defects, lead to an irregular cavity at the junction area between the spinal cord and the brachial plexus, (ii) subsequently inducing the excessive build-up of scar tissue, and, (iii) finally, impact the penetration of axons (*Carlstedt, 2008*; *Meng et al., 2019*, *2020*). Therefore, the use of appropriate filling materials with a drug load may be employed to promote nerve regeneration (*Baazil et al., 2015*; *Ding et al., 2019*). Poly (D, L-lactide-co-glycolide)-poly (ethylene glycol)-poly (D, L-lactide-co-glycolide) (PLGA-PEG-PLGA) is a temperature-sensitive hydrogel that has a wide range of applications in drug delivery owing to its good biocompatibility and safety. The fluidity and gel-forming properties of PLGA-PEG-PLGA allow it to effectively fill in tissue defects and sustain the delivery of drugs to the site of injury (*Ding et al., 2019*; *Zhang et al., 2014*). Further, studies have reported that combinations of PLGA-PEG-PLGA hydrogel with drugs can be more efficacious than drugs alone (*Ding et al., 2019*; *Huang et al., 2020*). Because treatment with drugs alone, if it is intravenous injection, after the metabolism of the various organs of the body, the actual dose of the drug reaching the injured site is very small, and it cannot even meet the needs of the injured site for treatment. Simply increasing the injected dose will not only affect the function of the other organs, and more drugs are wasted. If the drug is injected directly *in situ*, the fluidity of the drug liquid and the irregular shape of the injured site will cause the loss of most drugs and the utilization rate is low. But the drug-loaded hydrogel can provide excellent encapsulation

for NEP1-40 and avoid sudden release of it. Therefore, it will increase the efficacy of the drug (*Macaya & Spector, 2012*; *Shrestha et al., 2014*; *Zhang et al., 2011*). Therefore, it can facilitate nerve regeneration. The current study aimed to examine the effects of PLGA-PEG-PLGA hydrogel loaded with NEP1-40 on neural regeneration in an adult rat model of BPA and evaluate the mechanisms underlying regeneration.

# MATERIALS & METHODS

## Materials

PEG (Mn = 1,500 g/mol), D, L-Lactide (D, L-LA), glycolide (GA), Tin (II) 2-ethylhexanoate ($Sn(Oct)_2$), methyl thiazolyl tetrazolium, and quercetin were purchased from Sigma–Aldrich (Sigma, St Louis. MO, USA). NEP1-40 was purchased from TOCRIS (Bristol, UK). Sheep anti-rat choline acetyltransferase (ChAT) antibody was purchased from Millipore (Darmstadt, Germany), while rabbit anti-goat fluorescence 568 antibody and alpha-bungarotoxin (α-BTX) 594 antibody were purchased from Abcam (Cambridge, UK) and Invitrogen (Waltham, MA, USA), respectively. Fluoro-Gold (FG) was purchased from Fluorochrome, LLC. (Denver, CO, USA). All chemical reagents were used directly without further purification.

## Synthesis of the temperature-sensitive PLGA-PEG-PLGA hydrogel

As described previously (*Ci et al., 2014*; *Zhao, Guo & Ma, 2014*), firstly, 68 g of PEG, 115 g of D, L-LA, and 35 g of GA were dehydrated by vacuum drying for 48 h. Subsequently, 30 mg of Sn $(Oct)_2$ was added as a catalyst at 120 °C, and the reaction was allowed to continue for 48 h. The reaction product was sufficiently dissolved in 500 mL cold water at 4 °C, and the solution was heated to 80 °C for precipitation and purification. The purification process was repeated thrice, and the purified product was dried in a vacuum for later use.

## Characterization of PLGA-PEG-PLGA

Experimental data were recorded using a Bruker DMX500 spectrometer (Switzerland) at 500 MHz Deuterated chloroform ($CDCl_{3)}$ was used as a solvent, and tetramethylsilane was used as an internal standard. The sol–gel transition temperature of the PLGA-PEG-PLGA triblock copolymer was determined by the vial inversion test. Different quantities of PLGA-PEG-PLGA polymer were dissolved in phosphate-buffered saline (PBS), and the vial containing 0.5 mL of the polymer solution was stored for 15 min prior to the measurement. The temperature increment was set at 1 °C. The gel was successfully formed when the solution in the vial did not flow after being inverted for 30 s.

## Release profiles of NEP1-40 from PLGA-PEG-PLGA hydrogels

As described above, the optimal mass concentration of the PLGA-PEG-PLGA polymer was prepared. Then, NEP1-40 was dissolved in acetone and added to the PLGA-PEG-PLGA solution. Subsequently, the PLGA-PEG-PLGA polymer loaded with NEP1-40 was added to a test tube with an inner diameter of 16 mm and incubated at 37 °C for 10 min. After the hydrogels were formed, 5 mL of PBS were added to the top of the gels, and

the test tubes were placed at 37 °C with continuous shaking at 60 rpm. Subsequently, 1 mL of PBS was regularly aspirated and replenished every 2 days, with a total time of 14 days (*Nakamura et al., 2011*; *Xu et al., 2017*; *Zhai & Feng, 2019*). The amount of NEP1-40 in the aspirated PBS extract was measured using an ultraviolet spectrophotometer at 306 nm, and drug release profile was constructed on the basis of the obtained absorbance values.

## Experimental animals and groups

Animal procedures were conducted in accordance with the guidelines for the review and approval of the Animal Care and Use Committee of Jilin University (Ethical Approval No. 2018-119), and followed the Guide for the Care and Use of Laboratory Animals issued by the USA National Institutes of Health (*National & Promising, 2009*). Effort was made to minimize pain and the number of animals used in the experiments. The dose used for anesthesia is 45 mg/kg of 3% pentobarbital sodium (New Asia Pharmaceutical, Shanghai, China), and the dose used for euthanasia is 200 mg/kg of 3% pentobarbital sodium.

Forty healthy female adult Sprague–Dawley rats (weight, 200–250 g; age, 10–12 weeks) were provided by the Animal Experimental Center of Jilin University in China. The reasons why we choose female adult rats are as following: (1) female rats are less aggressive and have no sense of territory, so they are less aggressive to each other; while male rats have territorial awareness, they may attack each other, causing the experiment to fail; (2) after brachial plexus injury, some rats will be aggressive or eat limbs that do not feel sensation due to loss of sensation on one side. Females are less aggressive, so they are less likely to eat their limbs. All animals were caged in standard animal rooms at 22 °C with free access to food and water. The rats were randomized into the following four groups ($n$ = 10/group): control group, blank hydrogel group, NEP1-40 group, and NEP1-40-loaded hydrogel group. Experienced technicians for health and behavior monitored the animals daily during the protocol. The animals were weighed weekly. No rats displayed markers associated with death or poor prognosis of quality of life, or specific signs of severe suffering or distress, which would have led to early and immediate euthanasia.

## Model preparation and treatments

The rats were intraperitoneally anesthetized using 45 mg/kg of 3% pentobarbital sodium. The BPA model was established according to the methods described by *Gu et al. (2004)*. Briefly, the backs of rats were depilated, and the lamina and spinous processes from the fourth cervical spine (C4) to the second thoracic spine (T2) were exposed *via* the dorsal approach. The dorsal lamina of the right C5–C7 was opened followed by exposure and avulsion of the C5–C7 nerve roots. The C5 and C7 nerve roots (about 5 mm in length) were excised, and the distal ends of the nerve roots were retracted to prevent regeneration. The C6 nerve root was replanted to the spinal cord, and the muscles and skin were sutured after hemostasis. After surgery, the animals were maintained in a clean environment and administered an intramuscular injection of cefazolin sodium (50 mg/kg,

**Table 1 Grouping and treatments.**

| | Control group | Blank hydrogel group | NEP1-40 group | NEP1-40-loaded hydrogel group |
|---|---|---|---|---|
| Material loaded around the replanted C6 nerve root | (–) | 100 μL hydrogel | (–) | 100 μL, 0.5 mg/mL NEP1-40—loaded hydrogel |
| Intraperitoneal injection | PBS/DMSO | PBS/DMSO | 12.5 μg NEP1-40 | PBS/DMSO |

twice a day; Zhong-nuo Pharmaceutical) as anti-infection treatment for 3 consecutive days (*Birrell & Fuller, 2019*; *Kusaba, 2009*).

In the control group, the surgical incision was sutured directly, and a 100 μL mixture of PBS and dimethyl sulfoxide (DMSO) (83% PBS + 17% DMSO) was administered intraperitoneally once a day for 14 consecutive days (Table 1). In the blank hydrogel group, 100 μL of hydrogel was injected around the replanted C6 nerve root and held for 3 min. The incision was sutured until the hydrogel was stabilized. Postoperatively, a 100 μL mixture of PBS and DMSO (83% PBS + 17% DMSO) was intraperitoneally administered once a day for 14 consecutive days (Table 1). In the NEP1-40 group, the surgical incision was directly sutured, and 12.5 μg of NEP1-40 were dissolved in a 100 μL mixture of PBS and DMSO (83% PBS + 17% DMSO) and intraperitoneally injected once a day for 14 consecutive days (Table 1) (*Nakamura et al., 2011*; *Steward et al., 2008*; *Xu et al., 2017*). In the NEP1-40-loaded hydrogel group, 100 μL of 0.5 mg/mL NEP1-40-loaded hydrogel was injected around the replanted C6 nerve root and allowed to stabilize for 3 min. The incision was then sutured. Postoperatively, a 100 μL mixture of PBS and DMSO (83% PBS + 17% DMSO) was intraperitoneally administered once a day for 14 consecutive days (Table 1).

## Behavioural test

Between weeks 2 and 6 post-surgery, eight rats from each group were subjected to the Terzis grooming test (TGT) test to evaluate motor function of the affected upper limb (*Bertelli & Mira, 1993*; *So & Xu, 2015*). Two investigators who did not participate in model induction graded the motor function of the right upper limb. Divergences between assessments of the two investigators were independently resolved by a third investigator.

## Immunofluorescence assay and FG nerve retrograde labeling

At 6 weeks post operation, three rats from each group were randomly selected and sacrificed. After infusion with 4% paraformaldehyde, the spinal C6 segment was dissected and cut into 25-μm thick sections. One out of every four sections was incubated with the anti-ChAT (1:100; Millipore) antibody at 4 °C overnight. The following day, sections were incubated with the secondary antibody at 18–22 °C for 1.5 h (Alexa Fluor® 568 rabbit anti-goat IgG; Abcam). ChAT-positive spinal anterior horn motoneurons on the injured side and contralateral side were counted under a fluorescence microscope to assess the cell ratio of the injured side to the healthy side.

At 6 weeks post operation, three rats were randomly selected from each group. They were anesthetized and a transverse incision was made below the right clavicle. The pectoralis major and minor muscles were cut off to expose and dissociate the

musculocutaneous nerve of the right side. A glass needle was inserted 5 mm proximal to the nerve entry point of the musculocutaneous nerve into the biceps muscle, and 0.8 μL of FG was carefully injected into the musculocutaneous nerve with a microinjection pump. The needle was retained for 10 s until the FG was completely absorbed. The muscle and skin were sutured layer by layer. Postoperative anti-infection treatment was administered for 3 consecutive days. Subsequently, the rats were subjected to cardiac perfusion using 4% paraformaldehyde, and the C5–C8 segments were longitudinally cut into 25-μm thick sections. One out of every four sections was observed under a fluorescence microscope to assess the number of FG-labeled neurons.

### Immunofluorescence detection of muscle fiber diameter in the injured biceps and assessment of the amount and morphology of motor endplate (MEP) neurons in the injured biceps muscle by immunofluorescence

At 6 weeks post operation, four rats were randomly selected and sacrificed from each group and subjected to cardiac perfusion using 4% paraformaldehyde. The biceps muscles on the injured side were dissected, cut into 14-μm thick sections, and stained with hematoxylin-eosin. We randomly selected and photographed 50–100 muscle fibers from each biceps muscle under a light microscope. Image-J (National Institutes of Health, Bethesda, MD, USA) was used to measure the diameter of the muscle fiber, and the average value was calculated.

One out of every four sections was used to stain the MEP. The sections were incubated with α-BTX 594 antibody in 0.01 M PBS (α-BTX 594: 0.01M PBS = 1:400) for 30 min, and the morphology of the MEPs was observed under a microscope. At least 150 MEPs were randomly selected from each animal specimen and photographed. Image-J analysis software was used to measure the area of the MEPs.

### Statistical analyses

Graph-Pad Prism 5.0 cartography software (Graph-Pad Software, Inc., CA, USA) was used for mapping, and SPSS 15.0 software (SPSS, Chicago, IL, USA) was used for one-way analysis of variance. Measurement data are expressed as means ± standard deviation (SD). A pairwise comparison was performed using the Students *t*-test. Differences with a *P*-value of ≤ 0.05 were considered statistically significant.

## RESULTS AND DISCUSSION

### Material characterization and release profile of NEP1-40

PLGA-PEG-PLGA temperature-sensitive hydrogel can (i) fill defected tissues, (ii) restrict the ingrowth of some cell types, including fibroblasts and inflammatory cells, and (iii) prevent astrocyte proliferation, thereby minimizing the formation of scar tissue.
New tissues will completely fill the injured area following the absorbance, degradation, and disappearance of the scaffold (*Luo, Borgens & Shi, 2004*; *Rong et al., 2019*; *Zhang et al., 2014*). In this study, we first synthesized PLGA-PEG-PLGA hydrogels and determined their properties.

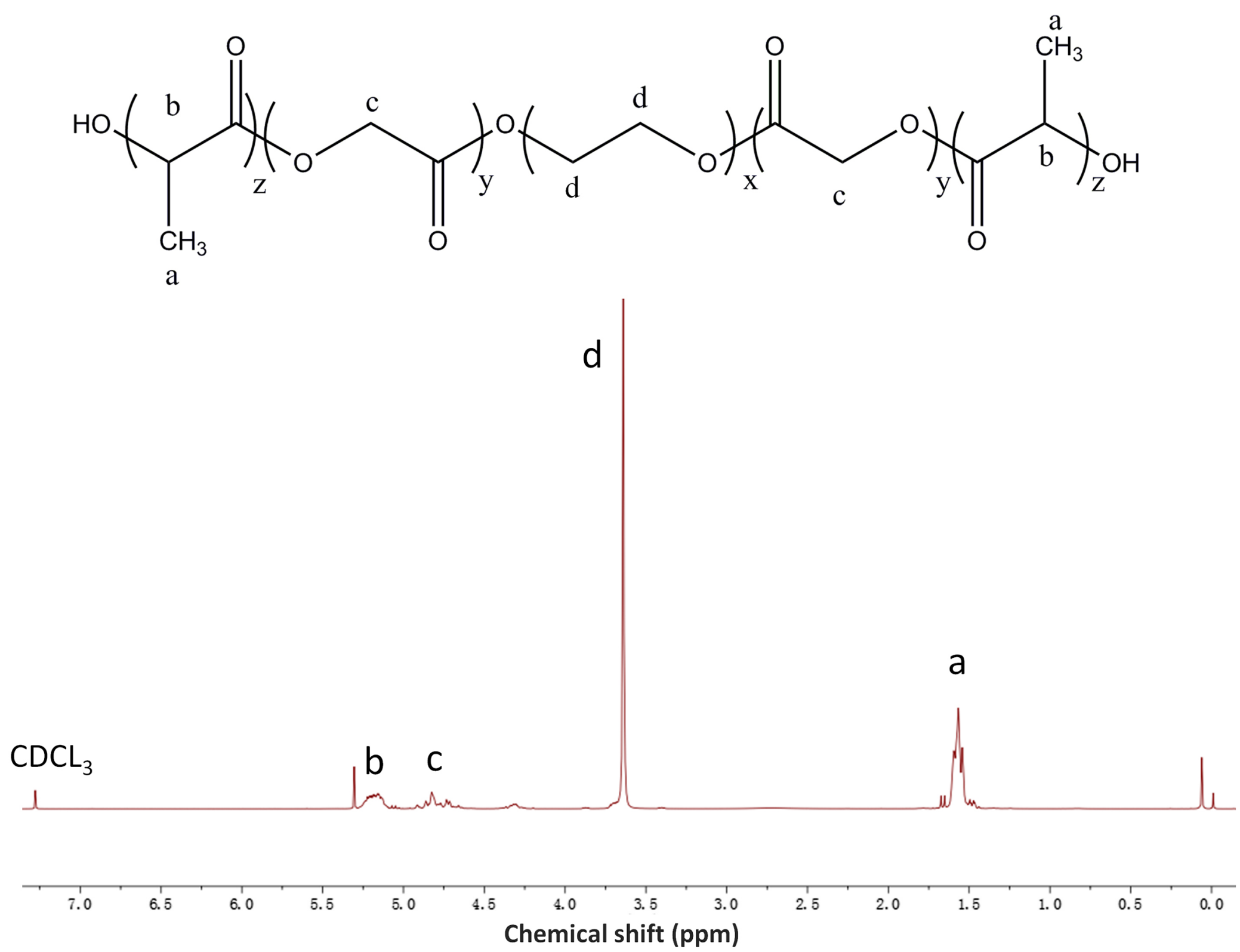

**Figure 1 The [1]H NMR spectrum of the PLGA-PEG-PLGA copolymer in $CDCl_3$.** The characteristic signals appearing at 5.2, 4.8, 3.5, and 1.5 ppm represent the CH of LA, $CH_2$ of GA, $CH_2$ of PEG, and $CH_3$ of LA, respectively.

The composition and structure of the PLGA-PEG-PLGA copolymer were determined using $^1$H NMR spectroscopy (Fig. 1). We observed a complicated split in these peaks owing to the random copolymerization of GA and LA, and the characteristic signals that appeared at 5.2, 4.8, 3.5, and 1.5 ppm represented the CH of LA, $CH_2$ of GA, $CH_2$ of PEG, and $CH_3$ of LA, respectively. Studies have revealed that there is a close association between the sol–gel transition of PLGA-PEG-PLGA hydrogel and its concentration (*Zhang et al., 2011*). In this study, the vial inversion test was performed to explore the sol–gel transition of the hydrogels (Fig. 2), and a typical sol–gel-precipitation transition was observed when the hydrogel concentration was 15–30%. Our findings revealed that a higher PLGA-PEG-PLGA concentration was associated with a higher gel-precipitation transition temperature and a lower sol-gel transition temperature, and a 30 wt% PLGA-PEG-PLGA hydrogel had a wider gel window. Therefore, the 30 wt% PLGA-PEG-PLGA hydrogel was selected for later use because of its appropriate sol–gel transition temperature.

We measured and plotted the drug release profile (Fig. 3) to observe the release of NEP1-40 from the PLGA-PEG-PLGA hydrogel. A rapid release of about 37% NEP1-40 was observed on day 1, and about 70% of NEP1-40 had been released within 14 days in a relatively smooth sustained-release trend. These findings indicate that the hydrogel can sustain drug release for at least 14 days.

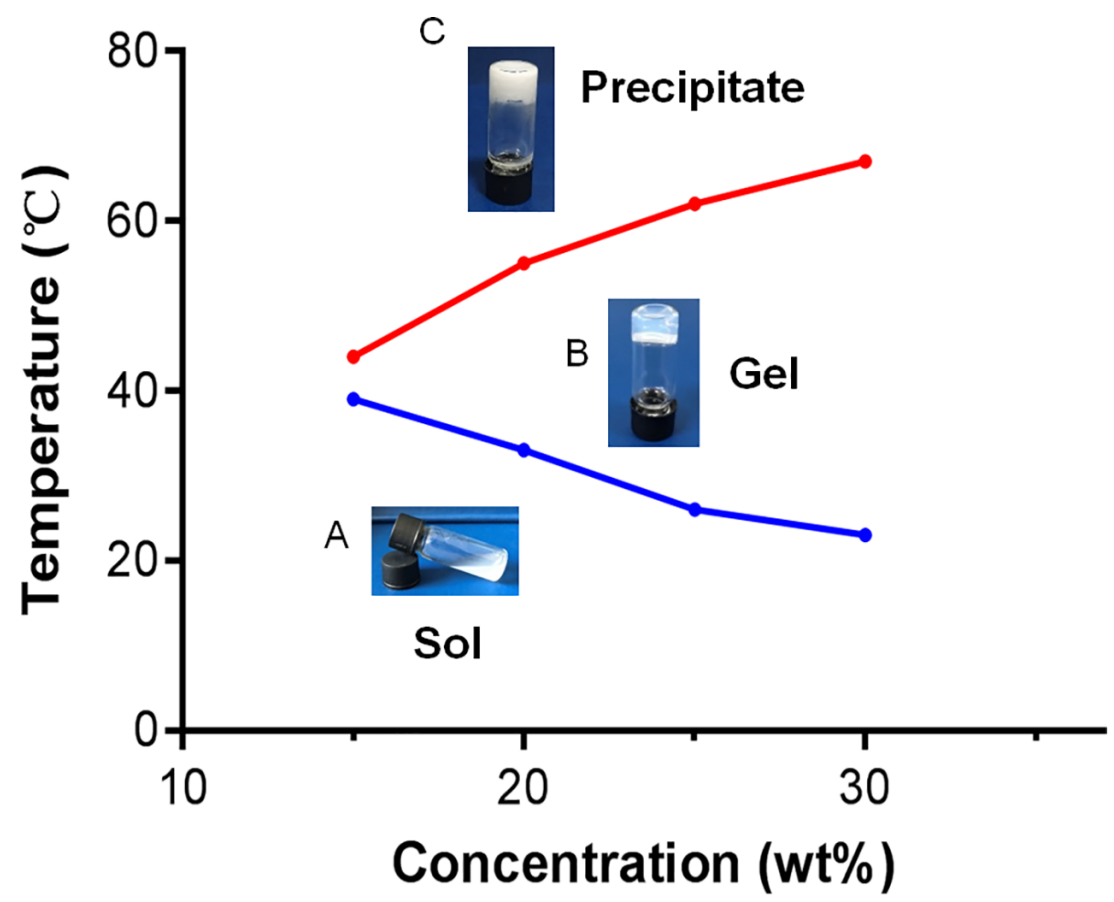

**Figure 2 Sol-to-gel transition temperature measurement of PLGA-PEG-PLGA copolymers determined using the vial inversion test and phase diagram of the sol-gel-precipitation transition as a function of PLGA-PEG-PLGA concentration.** (A) Copolymers in the sol state; (B) Copolymers in the gel state; (C) Copolymers in the precipitate state. PLGA-PEG-PLGA: poly (D, L-lactide-co-glycolide)-poly (ethylene glycol)-poly (D,L-lactide-co-glycolide).

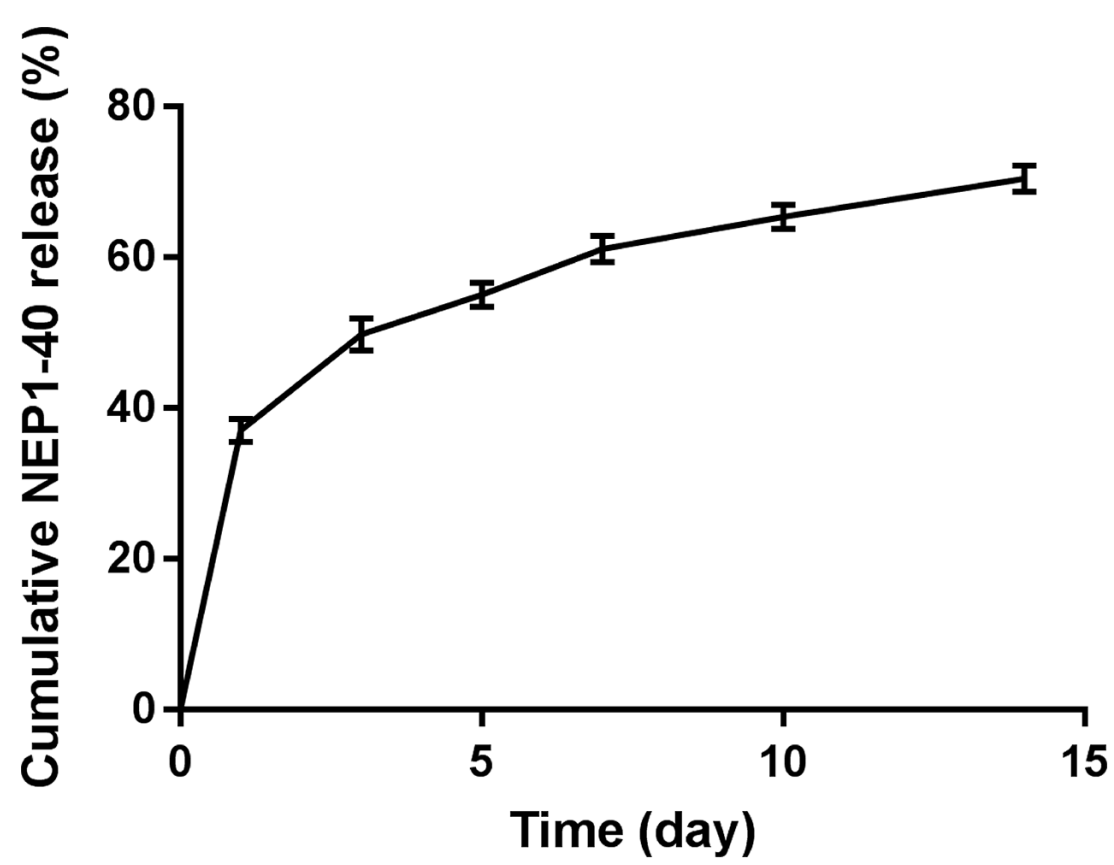

**Figure 3 Cumulative release profile of NEP1-40.** On day 1, about 37% NEP1-40 was released from the PLGA-PEG-PLGA hydrogel. During days 2–14, 70% of the NEP1-40 was steadily released. Data are presented as means ± standard errors ($n$ = 3).

### Treatment with the NEPI-40-loaded hydrogel sustained-release system significantly improves the function of biceps brachii on the injured side after nerve root replantation for BPA

*Steward et al. (2008)* showed that early application of NEP1-40 for 14 days can promote the recovery of spinal cord injury. And considering that the recovery period of brachial plexus injury (6–8 weeks) is shorter than that of spinal cord injury, and the time required for spinal cord axons to cross the spinal cord-peripheral nerve junction is shorter than that of spinal cord injury, therefore, we consider 14 days as the appropriate treatment time in our experiment (*Nakamura et al., 2011*; *Xu et al., 2017*; *Zhai & Feng, 2019*).

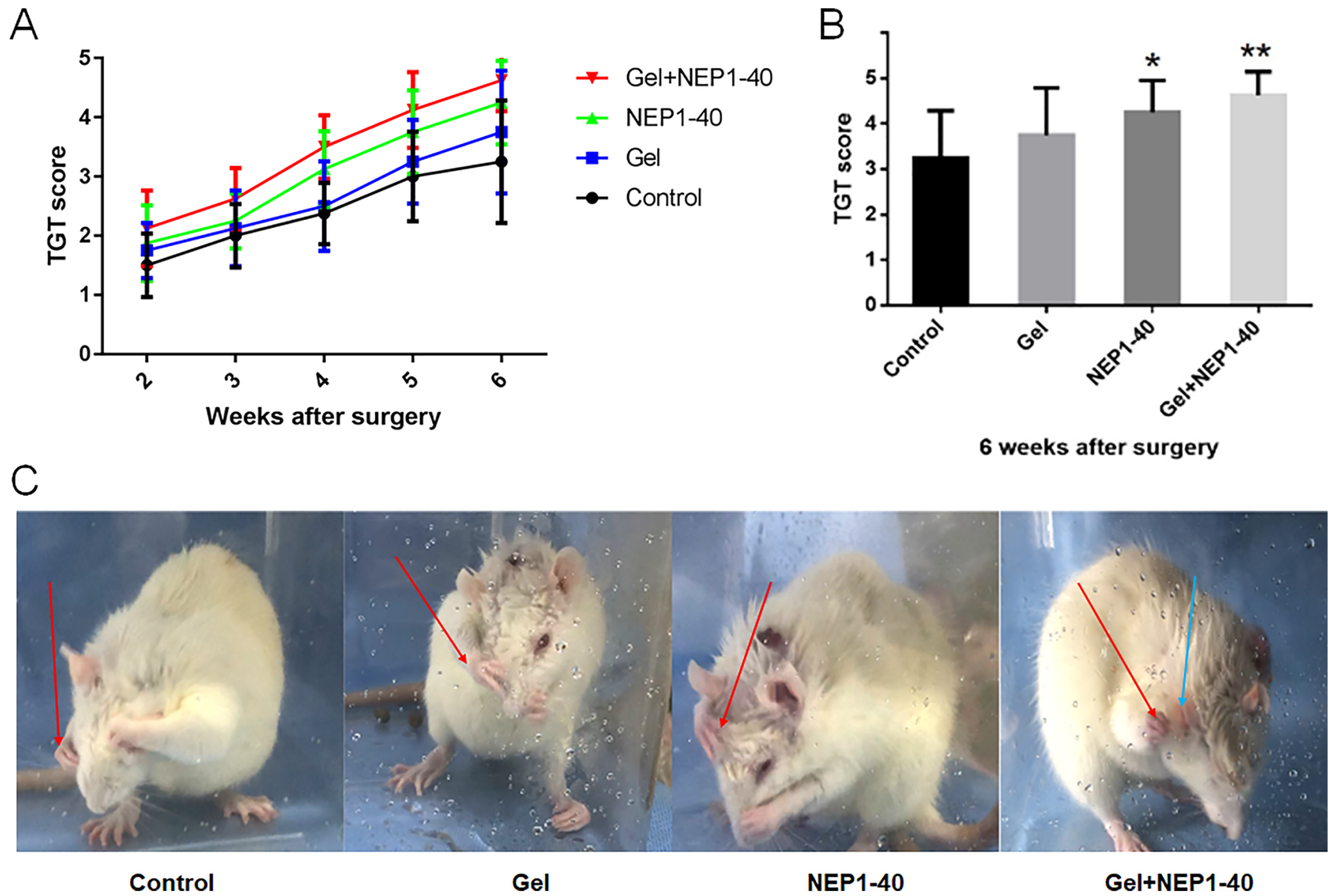

**Figure 4 TGT scores ($n$ = 8) of the control, blank hydrogel, NEP1-40, and NEPl-40-loaded hydrogel groups at 2–6 weeks post-surgery.** (A) TGT scores in each group at 2–6 weeks post-surgery. (B) TGT scores in each group at 6 weeks post-surgery ($^{*}P < 0.05$, $^{**}P < 0.01$). (C) Representative images of the rats at 6 weeks post-surgery. The rat in the control group did not touch its eyes with the affected upper limb (right), but the rat in the Gel+NEP1-40 group reached the ears with the affected upper limb (right). Arrow red: right upper limb; arrow blue: ear.

BPA model is confirmed by TGT as all the affected upper limb of animals found to receive grade of 0 on postoperative days 1 and 2 (*Bertelli & Mira, 1993*). This suggested that we successfully established the BPA model. Recovery of nerve function was the most important goal of our study. First, we investigated whether the NEP1-40 hydrogel sustained-release treatment improved the recovery of motor function in rats. We found that the NEP1-40 hydrogel sustained-release system promoted motor function of the upper limbs after nerve root replantation for BPA. At 2–6 weeks after injury, TGT scores were higher in the NEP1-40-loaded hydrogel group than those in the control, blank hydrogel, and NEP1-40 groups. At 2–3 weeks after injury, TGT scores were significantly higher in the NEP1-40-loaded hydrogel group than those in the other groups, indicating a rapid recovery of nerve function. At 4 weeks after injury, TGT scores were rapidly increased in the four groups, and this increase was faster in the NEP1-40 and the NEP1-40-loaded hydrogel groups than in the other groups (Fig. 4A). At 6 weeks post-surgery, the average TGT scores were 3.25, 3.75, 4.25, and 4.63 in the control, blank hydrogel, NEP1-40, and NEP1-40-loaded hydrogel groups, respectively (Figs. 4B and 4C). TGT scores were significantly higher in the NEP1-40 (Fig. 4B; $^{*}P < 0.05$) and NEP1-40-loaded hydrogel group (Fig. 4B; $^{**}P < 0.01$) than in the control group. TGT scores in the blank hydrogel group were also higher than those in the control group, but this difference was not obvious. Therefore, treatment with the NEP1-40-loaded hydrogel

sustained-release system significantly increased the TGT scores of the affected limbs, indicating that nerve function recovered well (*GrandPré, Li & Strittmatter, 2002*).

## Treatment with the NEPI-40-loaded hydrogel sustained-release system significantly increases the survival rate of anterior horn motoneurons and the number of functional motoneurons on the injured side

Neuronal survival is the basis of nerve regeneration. In this study, we observed the number of anterior horn motoneurons and functional motoneurons in the injured spinal cord. The spinal anterior horn motoneurons labeled by ChAT, a specific antibody which can help to distinguish between neuron and other tissues, were significantly bigger (their morphology was similar to normal) in the NEP1-40-loaded hydrogel group than in the other groups at 6 weeks post-surgery. Therefore, neuronal function was better preserved in the NEP1-40-loaded hydrogel group than in the remaining groups. The cell number ratios between the injured and healthy sides were 51.33%, 57.33%, 72.33%, and 75.67% (Fig. 5E; $^{*}P < 0.05$). Ratios were higher in the NEP1-40 and NEP1-40-loaded hydrogel groups than in the control group (Fig. 5E; $^{*}P < 0.05$).

FG nerve retrograde labeling of the C5–C8 segments revealed that the number of motoneurons in the spinal cord and their size increased significantly after treatment with the NEP1-40-loaded hydrogel sustained-release system (Figs. 5F–5J; $^{*}P < 0.05$, $^{**}P < 0.01$), indicating that the number of neurons involved in motor function was increased.
Due to the fact that the anterior horn motor neurons are limited to the anterior horn of the spinal cord, so the range is small. There are no positive cells in other places, which are blank. So, in order to display the cells more intuitively, parts with obvious cells are only put there (the original figures could be seen in the Supplemental File). Furthermore, the number of functional motoneurons was higher in the NEP1-40 treatment group than in the control treatment group (Fig. 5J; $^{*}P < 0.05$). The number of functional motoneurons was slightly but insignificantly increased in the blank hydrogel group as compared to the control group.

Regenerated axons need to pass through or bypass scar tissue in the injured area. Scar tissues contain various nerve growth inhibitory factors that make axonal regeneration tortuous and increase the probability of neural mismatch. Thus, the efficiency of axonal regeneration is decreased, and the difficulty of nerve regeneration is increased (*Cullheim, Carlstedt & Risling, 1999*; *Soderblom et al., 2013*). After BPA, nerve root replantation initiates axon regeneration and increases the survival rate of numerous neurons (*Gu et al., 2004*). Furthermore, axons grow rapidly, and the survival rate of neurons is greatly improved (*Li et al., 2015*). Our results revealed that NEPl-40 promoted axonal growth, thereby preventing necrosis of neurons. Therefore, NEP1-40 had a certain neuroprotective effect, providing a good foundation for nerve regeneration. In addition, we assessed the number of functional motoneurons: the number of FG-labeled neurons was significantly higher after treatment with the NEP1-40-loaded hydrogel sustained-release system than after control treatment (the control group) (Fig. 5J; $^{**}P < 0.01$) or the NEP1-40-alone treatment (the NEP1-40 group) (Fig. 5J, $^{*}P < 0.05$). This finding indicates that hydrogel

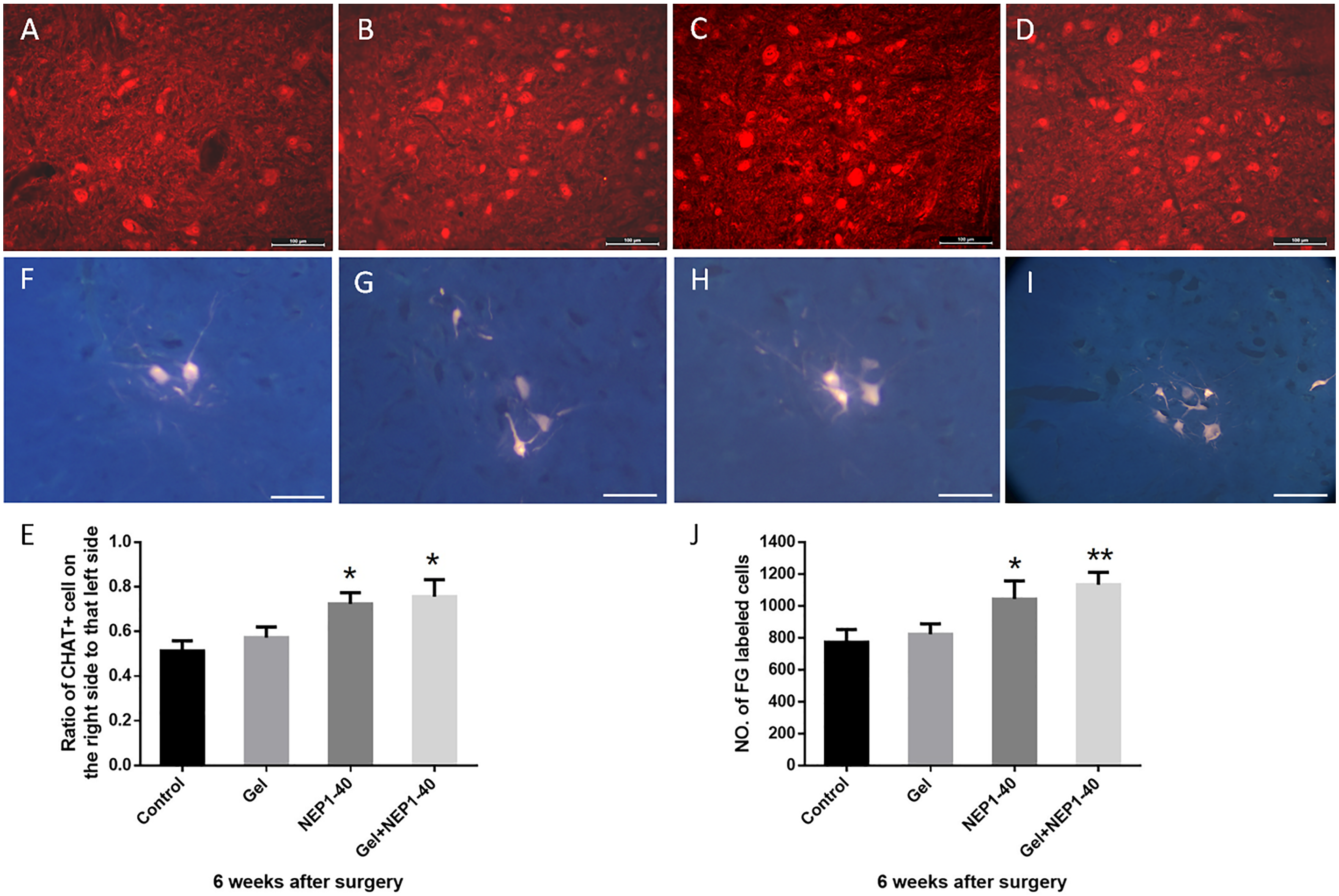

**Figure 5 Survival rates of anterior horn motoneurons and the number of FG-labeled motoneurons on the injured side C6 segment at 6 weeks post-surgery.** (A–D) ChAT (+) neurons (red) of the C6 spinal segment in the control (A), blank hydrogel (B), NEP1-40 (C), and NEPl-40-loaded hydrogel (D) groups, respectively (scale bar = 100 µm). (E) Survival rates of anterior horn motoneurons of the C6 segment ($n = 3$, $^{*}P < 0.05$). (F–I) FG nerve retrograde labeling of anterior horn motoneurons of the C5–C8 segments in the control (F), blank hydrogel (G), NEP1-40 (H), and NEPl-40-loaded hydrogel (I) groups, respectively ($n = 3$, scale bar = 100 µm). (J) The number of FG-labeled anterior horn motoneurons of the C5–C8 segments ($n = 3$, $^{*}P < 0.05$, $^{**}P < 0.01$).

and NEPl-40 treatment have synergistic effects, and more neurons participate in functional recovery, thereby promoting the restoration of nerve function.

## Treatment with the NEPI-40-loaded hydrogel sustained-release system can promote the regeneration of bicep muscle fibers and MEPs on the injured side

After BPA, denervation of muscle tissues leads to atrophy and dysfunction (*Duijnisveld et al., 2017*). Regeneration and growth of axons into target muscles allows for the recovery of atrophic muscles, fibrosis reversion of muscle fibers, and thickening of the diameter of muscle fibers to adapt to functional changes. MEPs in the muscles are concurrently re-innervated, and the MEP size is enlarged, recovering the original structure (*Ali et al., 2016*).

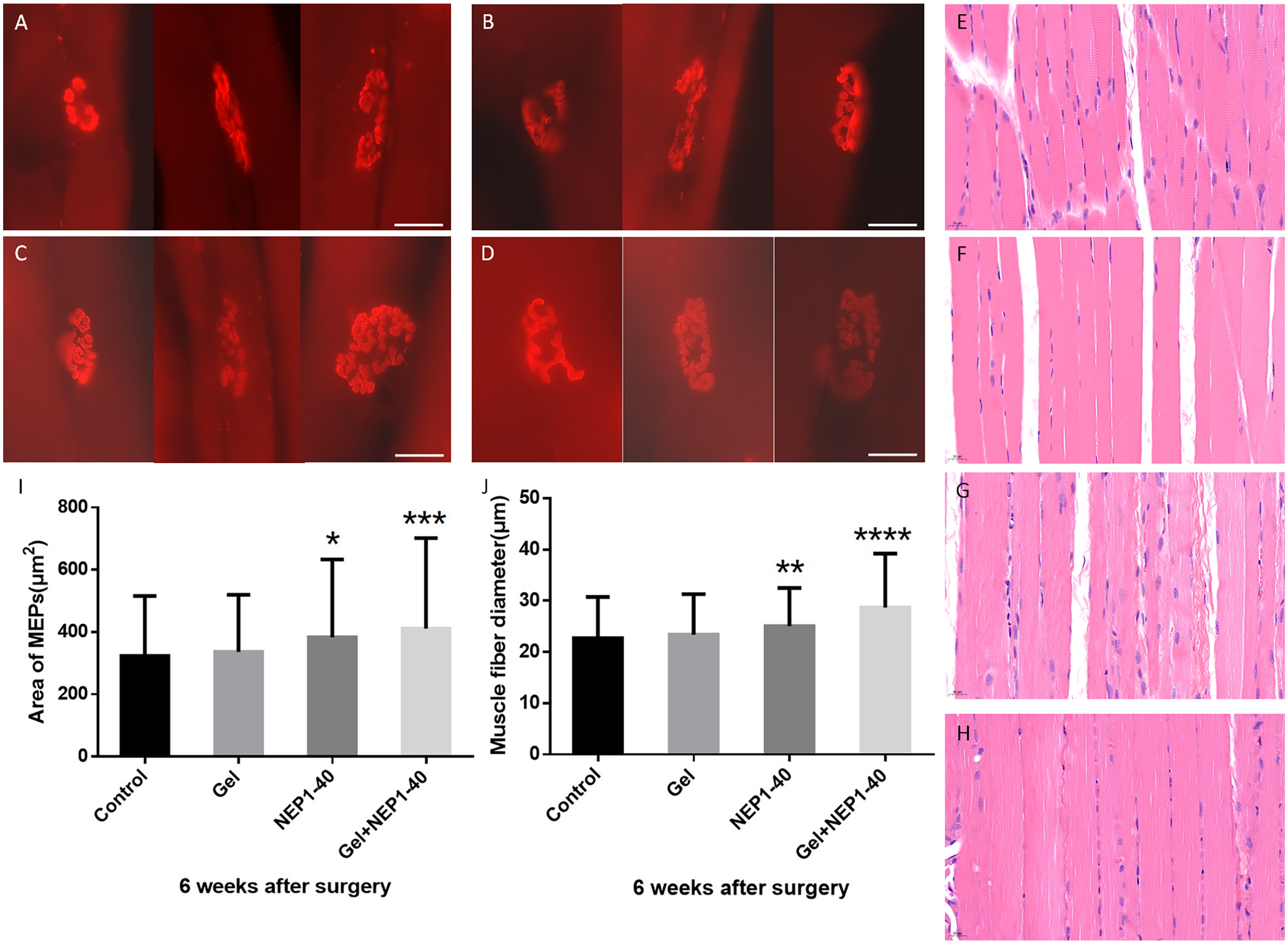

**Figure 6 Immunofluorescence staining of MEPs of the injured biceps brachii and hematoxylin-eosin staining of muscle fibers in the control (A), blank hydrogel (B), NEP1-40 (C), and NEPl-40-loaded hydrogel groups (D) at 6 weeks post-surgery.** A–D: MEPs of the injured biceps brachii in each group (red) ($n$ = 4, scale bar = 20 μm). E–H: Hematoxylin-eosin staining of the biceps brachii in the control (E), blank hydrogel (F), NEP1-40 (G), and NEPl-40-loaded hydrogel (H) groups, respectively ($n$ = 4, scale bar = 20 μm). (I) The area of MEPs of the biceps brachii on the injured side at 6 weeks post-surgery ($^{*}P < 0.05$, $^{**}P < 0.01$). (J) Muscle fiber diameter of the biceps brachii on the injured side at 6 weeks post-surgery ($^{***}P < 0.01$, $^{****}P < 0.0001$).

MEPs can be an important indicator of muscle function recovery after reinnervation (*Wang et al., 2018*). Measurements of the area of motor endplates in the injured biceps brachii at 6 weeks post-surgery revealed that MEP area increased more significantly after treatment with the NEP1-40-loaded hydrogel sustained-release system than with the other treatments (other groups). Additionally, MEP structure was clearer and more complex after treatment with the NEP1-40-loaded hydrogel sustained-release system than after other treatments. The area of MEPs in the NEP1-40-loaded hydrogel group was 27.3% higher than in the control group (Figs. 6A–6D, and 6I; $^{**}P < 0.01$), indicating that the NEP1-40-loaded hydrogel sustained-release system provides an anatomical basis for functional recovery. The size and complexity of MEPs were higher, and the area of MEPs

was 18.7% higher in the NEP1-40 treatment group than in the control group (Figs. 6A–6D, and 6I; $^{*}P < 0.05$), indicative of the effectiveness of NEP1-40 treatment.

Regenerated axons enter MEPs to trigger an increase in the volume of target muscle fibers, thereby leading to functional recovery. We performed hematoxylin-eosin staining of the muscle fibers of the injured biceps brachii at 6 weeks post-surgery and assessed fiber diameter. We found that muscle fiber diameter was the highest in the NEP1-40-loaded hydrogel group. Additionally, the muscle fiber diameter was 28.8% higher in the NEP1-40-loaded hydrogel group and 16% higher in the NEP1-40 group than in the control group (Figs. 6E–6H, and 6J; $^{****}P < 0.0001$, $^{**}P < 0.01$, respectively). The increase in muscle fiber diameter indicates a good recovery of function. However, muscle fiber diameter in the blank hydrogel group was only slightly higher than in the control group.

## CONCLUSION

Our findings indicate that the sustained release of NEP1-40 from the PLGA-PEG-PLGA hydrogel promotes nerve regeneration and recovery of motor function in BPA rats, promotes axonal growth, and protects neurons. The PLGA-PEG-PLGA hydrogel acts as a slow-release carrier and concurrently fills defects, thereby providing favorable conditions for nerve regeneration. Therefore, the combination of NEP1-40 and PLGA-PEG-PLGA hydrogel can promote axonal growth and recovery of nerve function. The present findings shed light on the potential application of the PLGA-PEG-PLGA hydrogel for the treatment of brachial plexus root avulsion.

### Funding

This study was supported by the Jilin Provincial Educational 135 Science and Technology Project (JJKH20190047KJ). The funders had no role in study design, data collection and analysis, decision to publish, or preparation of the manuscript.

### Grant Disclosures

The following grant information was disclosed by the authors:
Jilin Provincial Educational 135 Science and Technology Project: JJKH20190047KJ.

### Competing Interests

The authors declare that they have no competing interests.

### Author Contributions

- Wenlai Guo conceived and designed the experiments, analyzed the data, prepared figures and/or tables, authored or reviewed drafts of the paper, and approved the final draft.
- Bingbing Pei performed the experiments, prepared figures and/or tables, and approved the final draft.
- Zehui Li performed the experiments, authored or reviewed drafts of the paper, and approved the final draft.

- Xiao Lan Ou performed the experiments, prepared figures and/or tables, and approved the final draft.
- Tianwen Sun performed the experiments, analyzed the data, authored or reviewed drafts of the paper, and approved the final draft.
- Zhe Zhu conceived and designed the experiments, performed the experiments, prepared figures and/or tables, authored or reviewed drafts of the paper, and approved the final draft.

## Animal Ethics

The following information was supplied relating to ethical approvals (*i.e.*, approving body and any reference numbers):

This study was approved by the Research Ethics Committee of the Second Hospital of Jilin University (Ethical Approval No. 2018-119).

## Data Availability

The raw measurements are available in the Supplemental File.

## Supplemental Information

Supplemental information for this article can be found online at http://dx.doi.org/10.7717/peerj.12269#supplemental-information.

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
