# Peer review of "PLGA-PEG-PLGA hydrogel with NEP1-40 promotes the functional recovery of brachial plexus root avulsion in adult rats"

_PeerJ, doi:10.7717/peerj.12269_

## Round 0.1 · original submission · Major Revisions

Please address the concerns of all reviewers and revise your manuscript accordingly.

Reviewer 1 ·

Basic reporting

1-There were only a few spelling mistakes which could easily be corrected.
2-All the in-text citaitons have to be edited acording to journal's instructions.
3- Some parts of the introduction needs proper citations.
4- All the figures were uploaded in pdf format and their resolutions are very low. The author guidelines of the journal clearly indicates PNG and JPEG are accepted formats for figures and they should be 300 dpi in resubmission.
5-Some parts of the materials and methods section is actually belongs to results and discussion section. Reshaping is needed in this matter.
6-Discussion part should be improved.

Experimental design

1-The reseach question is well defined and answered.
2-Methods seems suffiecient. A few comments were made in this context which can be seen in the uploaded file.

Validity of the findings

I would like to re-evaluate this part after authors upload necessary images as supplementary documents.

Additional comments

Dear Author,
I’ve finished reviewing your manuscript on behalf of PeerJ. It was a pleasure for me to review this article. I believe that your manuscript could benefit some editing especially by improving the discussion part. When I checked the figures I saw that they were all in pdf format and the quality was very low. I also could not see any pictures in the supplementary files despite you’ve menioned it in the manuscript. I would kindly like to re-evaluate this manuscript after you make necessary corrections and upload proper images. You can see my detailed comments at the end of the attached file.
Best Regards

Annotated reviews are not available for download in order to protect the identity of reviewers who chose to remain anonymous.

Reviewer 2 ·

Basic reporting

The manuscript has a professional language and quality of the English and editing is good.

Literature references are included and sufficient background is provided.

Raw data is shared, article structure, figures and tables are good. But, Figures 5E, 5J, 6I and 6J may be bigger for easier reading.

The manuscript has relevant results to its hypotheses.

Experimental design

The study is in within aims and scope of the journal.

Research question is well defined and the method used is meaningful to fill the knowledge gap. It can be helpful to include in vitro literature in the references.

Investigation performed has good technical and ethical standard.

Methods are described with sufficient detail to replicate.

Validity of the findings

All data have been provided and they are well controlled and statistically sound. The authors may comment on the reason why the healthy side of the animals were not included as controls instead of a seperate control group. And the authors may also comment why they chose 6 weeks time point postoperative. They should include references for this time point and comment on neural regeneration regarding this time point.

Conclusions are well stated and linked to the research question. Authors may also include discussion on other behavioural methods for assessment and why they chose TGT method.

Reviewer 3 ·

Basic reporting

The article is clear and unambiguous. And professional English is used throughout. Literature references, background, and context are sufficient. Please revise/edit the following:

1. Line 80 needs to be revised as it looks like there is a missing statement. The line is inconsistent with the previous statement.

Experimental design

The experimental design is good and the research question is well defined. The investigation performed is well versed. I would like to see the following added/revised:

1. In line 177, include details or reference articles on the rationale justifying choosing reduced treatment time (14 days)

Validity of the findings

The release profiles have been well studied and documented. Conclusions are well stated and address the question raised. I suggest the authors could share more details on the concentration of NEP1-40 in the hydrogel would be appreciated- please also include a rationale, or cite references as to your decision to choose this concentration. The concentration and drug release profiles can be addressed in the conclusion, material preparation.

Additional comments

It is a well-detailed article that needs minor revisions. There is novelty in the study conducted, and the impact is well defined. I commend the authors for their study design and data set shared. The manuscript is clearly written in professional, unambiguous language. The minor revisions stated in the review need to be visited before Acceptance

Reviewer 4 ·

Basic reporting

Line 56, “associated protein-related ligands” should be “receptor”.

Line 59 should cite paper Nature. 2002 May 30;417(6888):547-51, Nogo-66 receptor antagonist peptide promotes axonal regeneration.

Line 62-64, cite reference specifically after each marker.

Line 175-179, please delete repetition.

Line 297 “in order to display the cells more intuitively, parts with obvious cells are only put there (the original figures could be seen in the supplementary file).” I don’t see the original pictures in supplementary file.

Fig.5A-D picture resolution is too low, but it looks like NEP1-40+hydrogel have less survived motoneurons than NEPI40. Also, E and J are hard to read.

Fig. 6 Lacks mice number in legend.

Paragraph starting from line 55 introduces Nogo signaling, but is poorly organized. Nomenclature should be clear and consistent. Please consider rewrite this paragraph.

Please discuss potential hydrogel toxicity if any or cite relative reference about its safety.

Authors mention “In our experiment, we used the drug NEP1-40, with the dosage and time (14 days) referring to relevant literature [1-3]” twice, please specify the dosage.

Please provide reason why female rats are selected.

Experimental design

Nogo has 3 variants. Is NgR1 expressed in the BPA injury site? Authors can cite a reference.

As authors mentioned, there are couple biomarkers are associated with Nogo signaling and are indicative of functional improve, such as GAP-43, MAP-2, APP. It would be necessary to look at these signaling in vivo to confirm NEP1-40’s protective effect from the mechanistic perspective.

Lacks a real control group which have no injury. Recovery by treatments need to be compared to no injury to get an idea how much is recovered.

Validity of the findings

For comparison between multiple groups (>2 groups), what statistic was performed, for example, on way or two way ANOVA followed by post hoc tests, please specify in Method. Students t-test is not appropriate.

Fig.4A, statistics of comparison between 4 groups at each time point can be added to the figure.

---

## Round 0.2 · Minor Revisions

Please address the remaining concerns of the revisers and revise manuscript accordingly.

Reviewer 1 ·

Basic reporting

1- The english language is clear.
2- Literature references and article structure is sufficiently edited after referree recommendations.
3- In text citations are properly corrected.
4- Newly uploaded figures are satisfactory.
5- Authors moved parts of the manuscript to the related section and made an effort to enrich the discussion part.
6- Authors should mention which picture represents which group in figure legends. (For ex in figure 5; CHAT (+) cells in control group (A), blank hydrogel group (B),.....)

Experimental design

It is not essential but it could be better if authors could give catalog numbers and dilutions of their primary and secondary antibodies. It could help other scientsists to design their studies and also would make the article matching with the repeataility principle for scientific research.

It would be better if authors could add calatog number for anti-Chat primary ab and rabbit abti-Goat secondary ab (and dilution) in this manner.

Validity of the findings

Supplementary data is well provided.
Results are well stated.

Additional comments

Dear author;
First of all I would like to offer my best wishes regarding your efforts on COVID-19 pandemic. I finished re-reviewing your manuscript on behalf of PeerJ and decided to recommend minor revisions. You can see my remaining concerns in the report.

Best Regards.

Reviewer 2 ·

Basic reporting

All revisions are done by the authors, after first round of review. So, not other comment.

Experimental design

All revisions are done by the authors, after first round of review. So, not other comment.

Validity of the findings

All revisions are done by the authors, after first round of review. So, not other comment.

Reviewer 3 ·

Basic reporting

The article is clear and unambiguous. And professional English is used throughout. Literature references, background, and context are sufficient. Image corrections have been made. The suggestions and questions raised by the reviewers have been sufficiently addressed and answered.

Experimental design

The article is clear and unambiguous. And professional English is used throughout. Literature references, background, and context are sufficient. The rationale for the 14-day period has been addressed.

Validity of the findings

The release profiles have been well studied and documented. Conclusions are well stated and address the question raised. The rationale for choosing concentration and dosage has been addressed by the authors.

Additional comments

It is a well-detailed article and major revisions and changes have been addressed. There is novelty in the study conducted, and the impact is well defined. I commend the authors for their study design and data set shared. The manuscript is clearly written in professional, unambiguous language.

Reviewer 4 ·

Basic reporting

1. Comment: Paragraph starting from line 55 introduces Nogo signaling, but is poorly organized. Nomenclature should be clear and consistent. Please consider rewrite this paragraph.

Response:
We are so sorry for the carelessness, and have conducted a whole modification regarding the problems you pointed out.

Second comment: I cannot localize any changes in this paragraph (the third paragraph of Introduction) in terms of reorganization/ any rewriting.

2. Comment: Authors mention “In our experiment, we used the drug NEP1-40, with the dosage and time (14 days) referring to relevant literature” twice, please specify the dosage.

Response: Thanks for your question. Please see the file attached here.
[1] Jing, Xu, Jian, et al. Comparison of RNAi NgR and NEP1-40 in Acting on Axonal Regeneration After Spinal Cord Injury in Rat Models. [J]. Molecular neurobiology, 2017.

Second comment: Thanks for the clarification. Please add your dose and administration route in method as this is a key information.

3. Comment: Please provide reason why female rats are selected.

Response: Thanks for your question. The main reasons are the following two:
(1) Female rats are less aggressive and have no sense of territory, so they are less aggressive to each other. Because male rats have territorial awareness, they may attack each other, causing the experiment to fail. (2) After brachial plexus injury, some rats will be aggressive or eat limbs that do not feel sensation due to loss of sensation on one side. Females are less aggressive, so they are less likely to eat their own limbs.

Second comment: Thank you for providing the reason. Please include this reason briefly in method when introducing the animal models.

Experimental design

1. Comment: As authors mentioned, there are couple biomarkers are associated with Nogo signaling and are indicative of functional improve, such as GAP-43, MAP-2, APP. It would be necessary to look at these signaling in vivo to confirm NEP1-40’s protective effect from the mechanistic perspective.

Response: Thanks for your advice. We have conducted in-depth research on the basis of existing research to explore the mechanism and molecular pathways of NEP1-40. We will report on related research in the follow-up, hoping to better explain the mechanism of NEP1-40.

Second comment: Thank you. Examination of molecular biomarkers In vivo are usually necessary to confirm the biomechanical function recovery, in addition to histology level examination (motor neuron numbers, morphology of motor endplate).

Validity of the findings

1. For comparison between multiple groups (>2 groups), what statistic was performed, for example, on way or two way ANOVA followed by post hoc tests, please specify in Method. Students t-test is not appropriate.

Response: Thanks for your question. The statistics we use in our manuscript are to first compare blank hydrogel group, NEP1-40 group, and NEPl-40-loaded hydrogel groups with the control group, and then compare NEP1-40 group, and NEPl-40-loaded hydrogel groups with blank hydrogel groups, and finally compare NEP1-40 group with NEPl-40-loaded hydrogel group, so there is no simultaneous comparison between the three groups as you mentioned.

Comment: Thank you. The pair of comparisons authors reposed is typical post hoc test post ANOVA. Unlike the Bonferroni, Tukey, Dunnett and Holm post hoc methods, a set of individual t tests does not correct for multiple comparisons. As a community, we tolerate a 5% false positive rate (reject null even when true), ie. true negative rate=1-0.05=0.95;
for 2 t-tests, true accept rate=0.95*0.95=0.9025;
for 3 t-tests, true accept rate=0.95*0.95*0.95=0.8574;
for 4 t-tests, true accept rate=0.95*0.95*0.95*0.95=0.8145, ie False positive rate=(1-0.8145)=0.1855, which means multiple t-test increase chance of false positive rate, way more than 5%. That's why correction for multiple comparisons is needed, or researchers will need to account biologically for multiple comparisons when they interpret the data.

Additional comments

I would more appreciate if authors could response by specifying the line number of the changes/additions in the revised manuscript.

---

## Round 0.3 · accepted · Accept

All critiques were adequately addressed and the manuscript was revised accordingly. Therefore I am pleased to accept your manuscript now.